# Direct observation of tensile-strain-induced nanoscale magnetic hardening

Deli Kong[1,2], András Kovács [1] ✉, Michalis Charilaou[3], Fengshan Zheng [1,4], Lihua Wang[5], Xiaodong Han [5] ✉ & Rafal E. Dunin-Borkowski [1]

Magnetoelasticity is the bond between magnetism and mechanics, but the intricate mechanisms via which magnetic states change due to mechanical strain remain poorly understood. Here, we provide direct nanoscale observations of how tensile strain modifies magnetic domains in a ferromagnetic Ni thin plate using in situ Fresnel defocus imaging, off-axis electron holography and a bimetallic deformation device. We present quantitative measurements of magnetic domain wall structure and its transformations as a function of strain. We observe the formation and dissociation of strain-induced periodic 180° magnetic domain walls perpendicular to the strain axis. The magnetization transformation exhibits stress-determined directional sensitivity and is reversible and tunable through the size of the nanostructure. In this work, we provide direct evidence for expressive and deterministic magnetic hardening in ferromagnetic nanostructures, while our experimental approach allows quantifiable local measurements of strain-induced changes in the magnetic states of nanomaterials.

Studies of the link between structure and magnetism in materials have a long history, dating back to the 19th century when J. Joule and E. Villari discovered magnetostriction and magnetoelasticity. Since then, stress and stress annealing have found applications in the sensing, control and enhancement of magnetic properties in a wide range of materials and devices[1], ranging from magnetic random-access memory[2–4] to energy harvesting[5–7] and biomedicine[8,9].

Strain and magnetostriction are two key parameters that influence the formation of anomalous magnetic properties in thin films, which lead to rotatable anisotropy and the formation of high-density stripe domains[10–13]. The ability to control the residual stress that gives rise to stress impedance effects has found applications in highly sensitive magnetic field sensors[14]. However, strain-induced effects on magnetism on the nanoscale remain poorly understood, primarily due to the experimental challenge of measuring and quantifying magnetic

fields in nanomaterials while at the same time allowing local control over strain. Such effects can be studied by observing the rearrangement of magnetic domains in the presence of strain while making measurements of magnetic domain wall width, which is highly sensitive to fundamental and induced magnetic properties. However, commonly-used magnetic imaging techniques are limited to observing surface magnetic states[5,15] or are unable to provide quantitative magnetic information. Magnetic imaging methods based on transmission electron microscopy (TEM), such as Fresnel defocus imaging (commonly referred to as Lorentz TEM), differential phase contrast (DPC) imaging[16] and off-axis electron holography (EH)[17–19] can be used to study magnetic domain walls, as well as their rearrangements in the presence of external stimuli, Fresnel defocus imaging has been used to observe the effect of elastic stress on magnetic solitons in chiral magnets upon cooling[20,21]. In addition to magnetic domain wall

[1]Ernst Ruska-Centre for Microscopy and Spectroscopy with Electrons and Peter Grünberg Institute, Forschungszentrum Jülich, 52428 Jülich, Germany. [2]School of Physics and Optoelectronics, Faculty of Science, Beijing University of Technology, 100124 Beijing, China. [3]Department of Physics, University of Louisiana at Lafayette, 70504 Lafayette, Louisiana, USA. [4]Spin-X Institute, Electron Microscopy Center, School of Physics and Optoelectronics, State Key Laboratory of Luminescent Materials and Devices, Guangdong-Hong-Kong-Macao Joint Laboratory of Optoelectronic and Magnetic Functional Materials, South China University of Technology, 511442 Guangzhou, China. [5]Institute of Microstructure and Properties of Advanced Materials, Beijing University of Technology, 100124 Beijing, China. ✉e-mail: a.kovacs@fz-juelich.de; xdhan@bjut.edu.cn

imaging, high-spatial-resolution magnetic imaging of the interplay between electromagnetic fields and strain in ferromagnetic nanostructures and judicious control of this phenomenon also promises to provide routes toward the functional sensing of strain fields in materials.

Here, we present high-resolution measurements of magnetoelastic coupling between tensile strain and magnetization in a single-crystalline Ni nanostructure recorded using in situ Fresnel defocus imaging and off-axis EH. We directly observe the modification of internal anisotropy fields via the formation of highly structured magnetic domain walls. We compare our results with micromagnetic simulations to quantify the strain-induced anisotropy field and use them to explain how strain can be used to control the susceptibility of thin ferromagnetic metals. As Ni has negative magnetostriction[22,23], strain induces orthogonal rotation of the magnetization. By using a bimetallic deformation device[24], Fresnel defocus imaging and off-axis

EH, we visualize magnetic texture changes, domain wall structure and dynamics quantitatively in real space during tensile straining of a single crystalline Ni sample. In order to assess the reproducibility of the results, we perform multiple tensile straining and release cycles up to the plastic deformation régime in magnetic-field-free conditions.

## Results

### In situ straining and magnetic imaging

Figure 1a shows a schematic diagram of the bimetallic deformation device, which was prepared on a standard Mo TEM half-grid. Figure 1b, c shows bright-field (BF) TEM images of a focused ion beam-prepared sample in unstrained and strained conditions, respectively. Details of sample fabrication, the experimental geometry and the magnetic imaging techniques are given in "Methods".

Direct observations of strain-induced effects on the magnetic domain state of the Ni sample were carried out by recording Fresnel

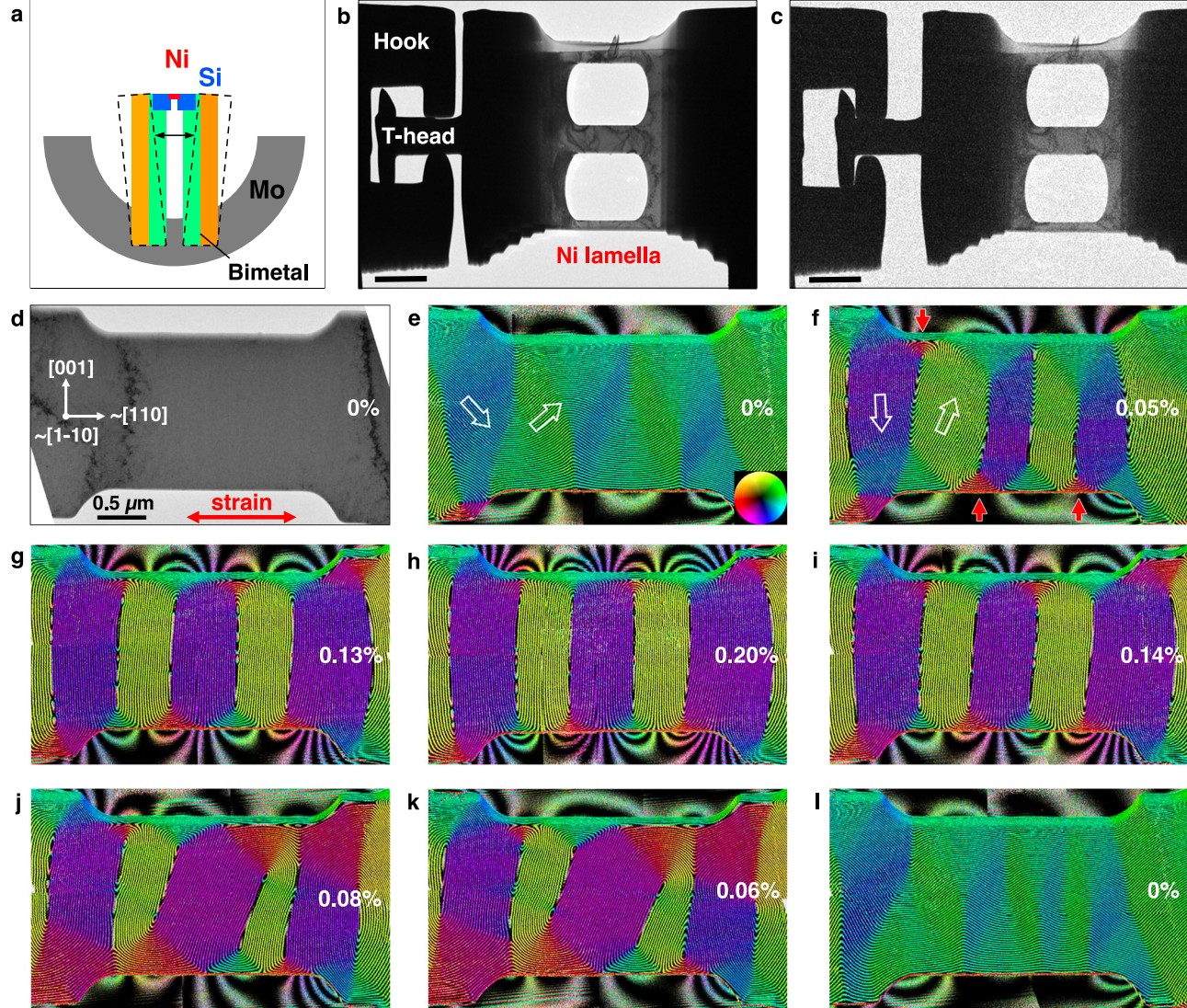

**Fig. 1 | Direct measurement of magnetostriction in a Ni nanostructure.**
**a** Schematic diagram of the bimetallic deformation device at room temperature and elevated temperature (dashed lines). **b**, **c** BF TEM images of the device in **b** unstrained and **c** strained conditions. The scale bar is 2 μm. In order to reduce the temperature effect, the distance between the hooks and the T-head was ~1 μm, enabling the straining experiment to be carried out below 60 °C. **d** BF TEM image of a single crystalline Ni nanostructure recorded before a second tensile cycle. The white arrows mark the crystallographic orientation of the sample. Strain was

applied in the horizontal direction (marked by a double-headed red arrow) at the levels indicated in each subsequent image. The error is estimated to be ±0.04%. **e–l** Real space magnetic induction maps recorded during a straining and release cycle using off-axis EH. The direction of the projected in-plane magnetic induction is visualized both according to the color wheel shown in (**e**) and using hollow white arrows. The contour spacing is 2π/3 radians. The small red arrows in (**f**) mark the positions of magnetic flux closure domains.

defocus images and off-axis electron holograms (Supplementary Fig. 1) during the application of tensile strain. Dynamical displacements of magnetic domain walls were observed as the strain was increased to 0.2% and then decreased to 0%. Supplementary Movie 1 shows the observed domain wall dynamics as a function of strain. The specifics of the magnetic transition were studied by visualizing the projected in-plane magnetic field inside and outside the specimen using off-axis EH.

The first tensile cycle experiment, which was used to determine the starting point for magnetic domain rearrangement, is shown in Supplementary Fig. 2. Maps of projected in-plane magnetic induction are shown in Fig. 1 during a second tensile cycle from 0 to 0.2% strain. Figure 1d shows a BF TEM image of the Ni sample before the second tensile cycle. The sample is single crystalline and shows no sign of structural defects, with the crystallographic [110] orientation nearly horizontal and parallel to the strain direction, which is marked with a double-headed red arrow.

Figure 1e reveals directly, in the form of magnetic field lines and colors, that the unstrained Ni sample comprises a magnetic domain structure of low angle (<90°) magnetic domain walls. The magnetic stray field is also recorded in the form of contours around the sample. Strikingly, when the tensile strain is increased to 0.05%, the magnetic field lines turn to lie perpendicular to the direction of the tensile strain. Figure 1f–h shows the evolution of the magnetic domain structure as the tensile strain is gradually increased to 0.2%. The low-angle magnetic domain walls transform into 180° domain walls, which are perpendicular to the strain axis. The 180° domain walls form a lattice. We also observe that the stray field intensifies outside the sample (Fig. 1e–h) despite the formation of flux closure domains as the system attempts to minimize the magnetostatic energy (marked with red arrows in Fig. 1f). The strength of the magnetic stray field outside the sample increases with strain as a result of increasing magnetostatic energy due to alignment in the domains and decreasing domain wall thickness, as discussed below.

The magnetic induction maps in Fig. 1 reveal the formation of periodic magnetic domains, which are separated by 180° magnetic domain walls perpendicular to the strain axis. In this experiment, the strain was released before the formation of any observable plastic deformation (i.e., before the formation of visible defects). On decreasing the strain to 0% (Fig. 1i–l), the magnetic domains transform back to a configuration that is comparable to the pristine condition, suggesting a reversible magnetoelastic process up to a strain of 0.20% (see also Supplementary Movie 1). We observed similar strain-induced magnetic responses in Ni samples with different crystallographic orientations, as shown in Supplementary Fig. 3.

## Magnetic domain wall structure

In order to quantify the relationship between strain and magnetic texture, the evolution of magnetic domain wall width with strain was determined from the derivative of the phase shift measured using off-axis EH[25,26] across two domains with a field rotation of 180° (Supplementary Fig. 4). The magnetic domain wall width was measured as a function of strain based on fits to the data, as shown using blue dots in Fig. 2a.

Strikingly, a significant decrease in magnetic domain wall width from 75 to 43 nm is observed when the tensile strain is increased from 0.05 to 0.20%. The magnetic domain wall width was determined for both straining and release cycles, confirming that the changes are reversible. The dramatic decrease in measured magnetic domain wall width $\delta$ with strain (by a factor of ~2) suggests that the ratio between exchange stiffness and effective anisotropy decreases by a factor of ~4 given that $\delta = \pi\sqrt{A/K}$, where $A$ is the exchange stiffness and $K$ is the magnetic anisotropy. This result, in turn, suggests that the strain-induced anisotropy is ~3 times larger than the shape anisotropy of the thin film. However, considering that the shape anisotropy is on the order of $10^5$ J/m$^3$, the latter interpretation is highly unlikely.

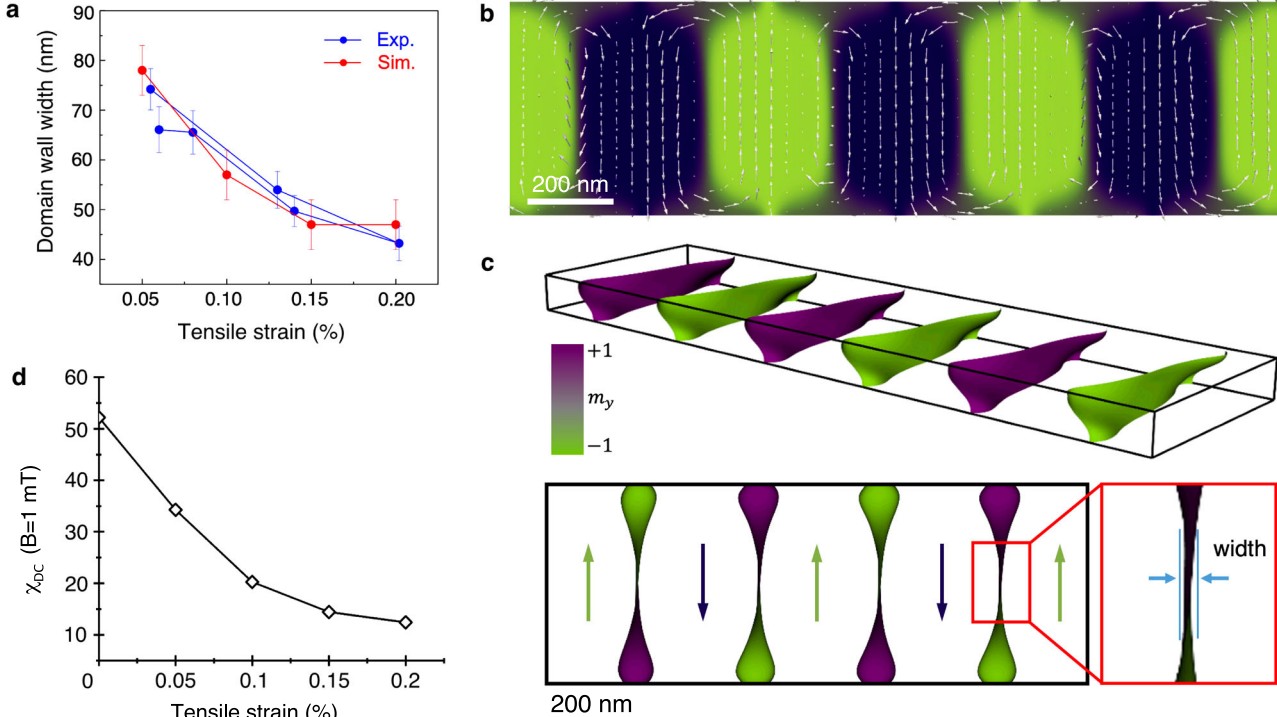

**Fig. 2 | Evolution of magnetic domain wall width with strain in a Ni nanostructure. a** 180° magnetic domain wall width measured (Exp.) as a function of tensile strain during the second tensile cycle using off-axis EH compared with micromagnetic simulations (Sim.). **b** Representative micromagnetic simulation of Ni in the presence of 0.20% strain. **c** Micromagnetic simulation of magnetic domain wall iso-surfaces, showing the three-dimensional shapes of the walls, which are twisted from edge to edge. With increasing strain, both twisting of the domain walls and the average magnetic domain wall thickness decrease. **d** Simulated DC magnetic susceptibility of the Ni nanostructure plotted as a function of tensile strain.

Instead, the apparent magnetic domain wall sharpening can be explained by considering the variation of the magnetic domain wall shape within the sample, considering that "*The domain wall is a three-dimensional object. The results obtained on thin films might be misleading*", as stated by Arrott[27]. This statement inspired a deeper study of the effect of strain on the magnetic state and an inspection of the magnetization arrangement of the 180° magnetic domain walls in three dimensions (3D) using micromagnetic simulations. The strain-induced micromagnetic state was simulated using a model that was based on the structure and dimensions of the Ni sample derived from imaging data and from a measurement of the sample thickness using electron energy-loss spectroscopy (Supplementary Fig. 1). In the model, the sample had dimensions (length × width × thickness) of $2000 \times 500 \times 100 \ nm^3$, while each simulation cell had a volume of $2 \times 2 \times 2 \ nm^3$. The total energy density included contributions from ferromagnetic exchange, cubic magnetocrystalline anisotropy and long-range dipole-dipole interactions (magnetostatics). Stress in the Ni film was incorporated in the simulations in the form of easy-plane anisotropy[28] perpendicular to the strain direction, i.e., normal to the [110] crystallographic axis. Strain-induced anisotropy ($K_s$) can be written in the form

$$K_s = -\frac{3}{2}\frac{Y}{(1+\mu)}\lambda\varepsilon$$

where $Y = 200$ GPa is the Young's modulus, $\mu = 0.3$ is the Poisson ratio, $\lambda_{110} = -31 \times 10^{-6}$ is the magnetostriction coefficient and $\varepsilon$ is the strain. The calculated value of strain-induced anisotropy is predicted to be between ~4 and ~15 kJ/m³ for strains of between 0.05 and 0.20%, respectively. The corresponding changes in magnetic state are summarized in Supplementary Fig. 5.

As stated above, these values of strain-induced anisotropy do not explain the reduction in magnetic domain wall width that is observed experimentally in Fig. 2a. However, the simulated magnetic domain wall structures capture the intricate details of the underlying strain-induced effects (Fig. 2b, c). Figure 2b shows the magnetic state in the presence of 0.20% strain in plan-view geometry. Even though the

magnetic field directions align perpendicular to the strain direction in the same way as in the experimental results shown in Fig. 1, analysis of the 3D magnetization distribution in the simulations reveals that the iso-surface of the wall is twisted from edge to edge, even at a relatively low level of strain, thereby appearing wider in projected images than its true thickness. With increasing strain, twisting of the magnetic domain walls scales down, leading to edge-on-views in the [1-10] direction, as shown in Fig. 2c. The projected magnetic domain wall width measurements from the simulation results are plotted in Fig. 2a and are in excellent agreement with the experimental observations.

The qualitative and quantitative agreement between the experiments and simulations allows the simulations to be used to predict further properties of strained Ni nanostructures and their response to external magnetic fields that cannot be measured directly. Figure 2d shows micromagnetic simulations of DC magnetic susceptibility, which is plotted as a function of strain predicted by simulating hysteresis curves of the same Ni sample in the presence of different values of strain. The strong decrease in susceptibility with tensile strain indicates a substantial magnetic hardening of the Ni nanostructures. The fact that the response of thin ferromagnetic films to external fields can be tuned by mechanical strain is of great importance for a wide range of applications, in part those that involve sensing of strain.

## Plastic deformation

In order to obtain insight into how the magnetic state evolves in the non-elastic regime, additional experiments were performed, where the sample was strained until evidence of plastic deformation was observed, as shown in Fig. 3. As the strain was increased gradually, the magnetization again rotated in a direction perpendicular to the strain (Fig. 3a, b), similar to the behavior shown in Fig. 1. However, once the elastic limit was exceeded at a strain of ~0.35% (Fig. 3c), stacking fault formation was identified, as confirmed by the BF TEM image shown as an inset to Fig. 3c.

The stacking faults had no major effect on the magnetic structure during the release process until the strain had been removed completely, as shown in Fig. 3d. Interestingly, at 0% strain, the magnetic domain structure was different after plastic deformation from that observed for the elastic cycles (Figs. 1d and 3a), with a nearly uniform in-plane magnetic field and the 90° domain walls eliminated. Despite

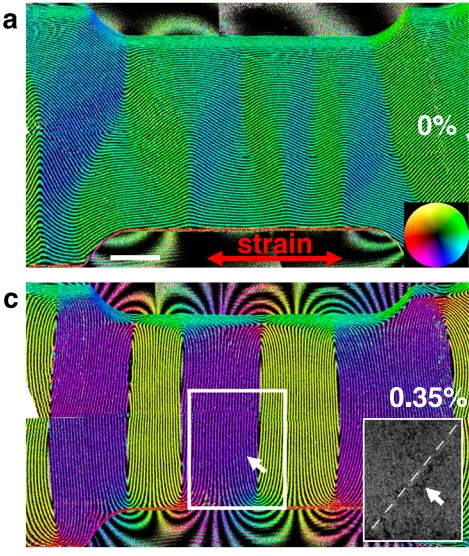
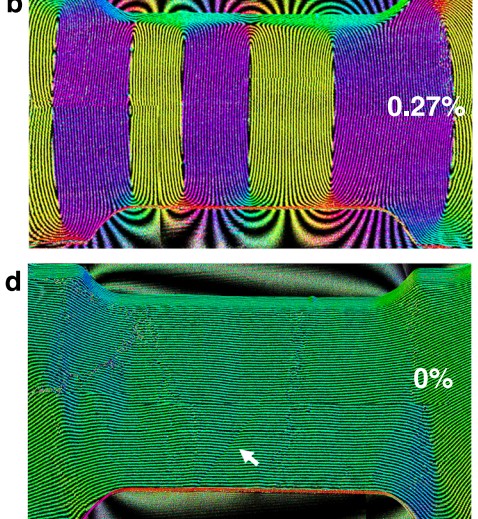

**Fig. 3 | Influence of plastic deformation on magnetic texture. a–c** Magnetic induction maps of the Ni nanostructure recorded before and after applying tensile strains of **b** 0.27% and **c** 0.35%, at which stacking fault formation was observed. The strain direction is marked by a double-headed red arrow. The inset in (**c**) is a BF TEM image corresponding to the white rectangular area. The stacking fault is marked by a dashed line and arrows in (**c**, **d**). **d** Magnetic structure after the stress was released. The direction of the projected in-plane magnetic field is visualized according to the color wheel shown in (**a**). The scale bar is 500 nm.

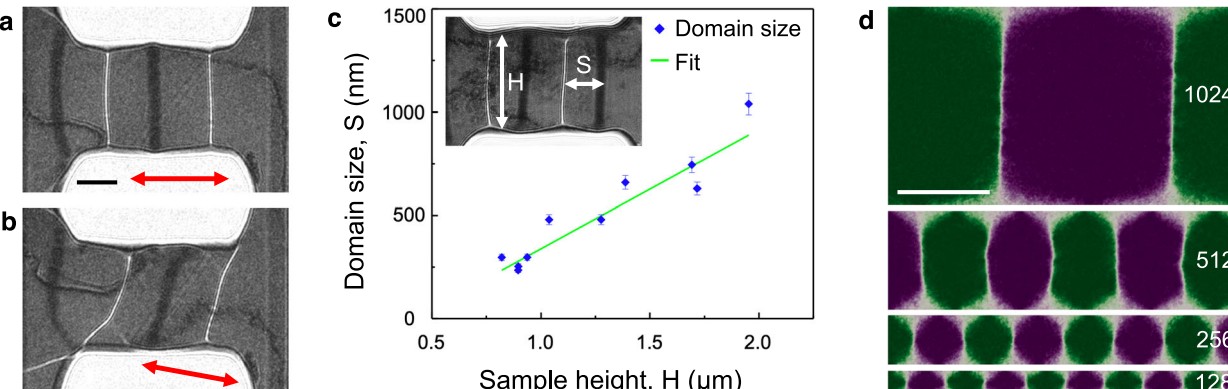

**Fig. 4 | Characteristic magnetic property changes in a strained Ni nanostructure. a, b** Fresnel defocus images of the Ni nanostructure, demonstrating directional control of magnetization by stress. The double-headed arrow marks the applied strain direction. **c** Effect of the height of the Ni nanostructure (H) on magnetic domain size (S) at fixed tensile strain (0.20 ± 0.05%). The linear fit is S = 0.58H-0.24, with an adjusted R-squared value of 0.85. **d** Micromagnetic simulations of Ni nanostructures of varying height at 0.20% strain showing changes in magnetic domain density. The scale bars in (**a**) and (**d**) are 500 nm.

the observed lack of reversibility of the magnetic state, when the strain was re-applied, the magnetic state returned to the same configuration as before, with domain walls perpendicular to the strain direction. These observations reveal the robustness of the magnetoelastic effect even when inverse magnetostriction is followed beyond the elastic limit in a nanostructure. Another straining cycle was conducted on the lamella following stacking fault formation, as shown in Supplementary Movie 2. The dynamic behavior of the magnetic domain walls under straining and release was observed to be similar to that for the elastic case.

### Effect of strain direction

The directional dependence of magnetization control via stress was tested by changing the strain direction relative to the film geometry. When the strain was parallel to the long horizontal side of the film, as in the original experiments, the directions of the in-plane magnetization and the magnetic domain walls were perpendicular to the strain direction, as shown in the form of a Fresnel defocus image in Fig. 4a. Directional control was achieved by manipulating the strength of the connection between the hook and the bimetallic deformation device. When the strain direction was changed by 9°, the magnetization direction also changed, as shown in Fig. 4b and Supplementary Movie 3, reconfirming the possibility of directional control of magnetization by strain. Furthermore, both our experimental observations and corresponding micromagnetic simulations revealed a correlation between the height of the Ni nanostructures and the sizes of the magnetic domains under tensile conditions. Figure 4c shows the dependence of magnetic domain size (defined to be the distance between two 180° domain walls) on sample height (see also Supplementary Fig. 6). The observed dependence is linear as a result of a balance between magnetostatic energy (which scales with volume) and magnetic domain wall energy (which scales with domain wall cross-sectional area).

### Discussion

We have combined experimental observations with micromagnetic simulations of magnetoelastic coupling between tensile strain and magnetic state to show that the application of strain to a single-crystalline ferromagnetic Ni nanostructure induces an easy-plane magnetic anisotropy, which results in rotation of the magnetization in a direction perpendicular to the strain axis. The magnetic state comprises periodic domain walls, which have a three-dimensional character and a spacing that depends on the dimensions of the sample. The pristine and unloaded magnetic states are highly similar when the system stays within the elastic regime, while they are

significantly different once stacking faults form in the plastic regime. The results provide an estimate of the stress that is required to introduce planar defects into a Ni nanostructure, which can be used to study the influence of planar defects on magnetic states in detail in future studies.

Our results demonstrate directly that a change in inter-atomic spacing associated with strain induces a substantial anisotropy that can lead to a transformative change in the magnetic state and magnetic hardening of a nanoscale sample. As the strain-induced magnetic state responds differently to an external magnetic field than a strain-free state, the susceptibility also depends on the strain. Similarly, charge transport is expected to depend on strain because anisotropic magnetoresistance depends on the angle between the current and the magnetization[29]. Our unique local observations of tunability between strain and magnetism suggest that nanoscale ferromagnetic films can be used as sensors of directional mechanical strain, either by measuring their magnetic susceptibility or via transport measurements in miniature devices.

### Methods

#### In situ mechanical straining transmission electron microscopy
In situ TEM was conducted using a home-built bimetallic deformation device developed in the group of Prof. Xiaodong Han and implemented in a Gatan Model 652 double-tilt heating holder in a transmission electron microscope dedicated to magnetic imaging. The bimetallic deformation device is made from two thermally-actuated bimetallic strips, which are fixed in opposing positions on a half TEM Mo ring using epoxy resin, as shown in Fig. 1a. Each bimetallic strip is made from layers of two different materials that have a large mismatch in their thermal expansion coefficients, in order to achieve a significant deflection at a relatively low operation temperature (<60 °C). The length of each strip is ~2 mm. A Si slice is fixed at the free end of each strip using epoxy resin. The distance between the two Si slices is ~30 μm. The Ni sample was prepared by using focused Ga ion beam sputtering in a dual beam scanning electron microscope (Thermo-Fisher Helios 600) and positioned between the two Si slices. A T-head, hooks and an electron-transparent region were made from a Ni single crystal, as shown in Fig. 1b. In a conventional TEM heating holder, the bimetallic deformation device can be heated moderately (<60 °C). The bimetallic strips bend in opposite directions with increasing temperature (Fig. 1b, c), causing the hooks to catch the T-head (Fig. 1c) and realizing an approximate uniaxial tensile test of the Ni lamella. The strain level is estimated as described in Supplementary Fig. 7. The slightly elevated temperature (<60 °C) has only a minor impact on the magnetism of Ni, as shown in Supplementary Fig. 8.

**Magnetic induction mapping using off-axis electron holography**
The projected in-plane magnetic induction in the Ni sample during the straining experiment was visualized and quantified using off-axis electron holography. Electron holograms were recorded using a spherical aberration corrected TEM (ThermoFisher (FEI) Titan 60-300) at 300 kV. The microscope was operated in aberration-corrected Lorentz mode with the sample in magnetic-field-free conditions. Fresnel defocus images and electron holograms were recorded on a direct electron counting detector (Gatan K2 IS) with 4k x 4k pixels. The typical biprism voltage was 70 V, which corresponded to a holographic interference fringe spacing of 4.46 nm and holographic interference fringe contrast in vacuum of ~30%. Image analysis was performed using Gatan Microscopy Suite and HoloWorks software. Supplementary Fig. 1 shows the processing steps that were used to reconstruct the projected in-plane magnetic induction. Typically, electron holograms were recorded from both the Ni sample and vacuum (Supplementary Fig. 1a, b). The total phase shift information (Supplementary Fig. 1c) was extracted using a standard Fourier transform method. The total phase shift provides information about local variations in both electrostatic and magnetic potential. Since the lamella is a single crystal and has negligible thickness variations (Supplementary Fig. 1e), it was assumed that the electrostatic contribution to the signal is constant and that any phase variations away from the sample edge are magnetic in origin. Magnetic induction maps were obtained by adding contours and colors to the phase images, as shown in Supplementary Fig. 1d.

**180° domain wall width measurement**
Supplementary Fig. 4 shows the procedure used for 180° domain wall width measurement. The position of a chosen 180° domain wall could be located accurately from a magnetic induction map of the lamella, as shown by a red rectangle in Supplementary Fig. 4a. The phase shift across the 180° domain wall at this position could be measured from the original phase image, as shown by a red rectangle in Supplementary Fig. 4b. The width of the 180° domain wall was determined from the differential of the phase shift, as shown by a black curve in Supplementary Fig. 4c. Nonlinear curve fitting (red line) to the differential of the phase shift was performed using the equation

$$y = \pm a \times \tanh\left(\frac{\pi \times (x - x_0)}{w}\right),$$

where $w$ is the width of the 180° domain wall, $a$ is an amplitude and $x_O$ is an offset that can be obtained from the fit. The domain wall width obtained from this particular measurement was $53.99 \pm 3.72$ nm.

**Micromagnetic simulations**
The simulation system had dimensions of $2000 \times 500 \times 100$ nm$^3$. Each cell had dimensions of $2 \times 2 \times 2$ nm$^3$. The total energy density $E$ of the Ni film with cubic symmetry was defined as

$$E = A \sum_i (\nabla m_i)^2 - M_s \boldsymbol{B}_{ex} \cdot \boldsymbol{m} + K_1(\alpha_x^2 \alpha_y^2 + \alpha_x^2 \alpha_z^2 + \alpha_y^2 \alpha_z^2) - \frac{1}{2} M_s \boldsymbol{B}_d \cdot \boldsymbol{m} + K_s(m_z)^2$$

where $A = 8.6 \times 10^{-12}$ J/m is the ferromagnetic exchange stiffness, $m_i$ is the $i^{th}$ component of the unit vector $\mathbf{m} = \mathbf{M}/M_s$ of the local magnetic moment, $M_s = 4.8 \times 10^5$ A/m is the saturation magnetization of Ni, $K_1 = -5000$ J/m$^3$ is the first order cubic anisotropy constant, $\alpha_i$ is the directional cosine with respect to the crystal axis, $\mathbf{B}_{ex}$ is the external field vector, $\mathbf{B}_d$ is the local field vector due to magnetostatic dipole-dipole interactions, $K_s$ is the strain-induced easy-plane anisotropy and $m_z$ is a vector parallel to the strain direction.

The equilibrium magnetic state of the sample was found by numerically solving the Landau–Lifshitz–Gilbert (LLG) equation

$$\frac{\partial \boldsymbol{m}}{\partial t} = -\gamma \boldsymbol{m} \times \boldsymbol{B}_{eff} + \alpha\left(\boldsymbol{m} \times \frac{\partial \boldsymbol{m}}{\partial t}\right)$$

where $\gamma = g\mu_B/\hbar$ is the gyromagnetic ratio with the Landé factor (g), the Bohr magneton ($\mu_B$) and the reduced Planck constant ($\hbar$). The dimensionless parameter α is a measure of the Gilbert damping, and the effective field in the material $\mathbf{B}_{eff} = \partial_{\mathbf{m}} E/M_s$ depends on internal and external fields. The LLG equation was integrated numerically with the software Mumax3[30].

Supplementary Fig. 5 shows a summary of the micromagnetic results in the form of contour plots of the local magnetization for the Ni film under different strains. In Supplementary Fig. 5b, c, the shaded area is a guide to the eye of the narrowing of the domain wall with increasing strain-induced anisotropy; the region where the stray field closes becomes progressively smaller.

## Data availability
All data in the main text or the supplementary information are available from the corresponding authors upon request.

## Code availability
Micromagnetic code used in this work is available upon request from M.C.

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

## Acknowledgements

The authors are grateful to C. Thomas and M. Feuerbacher for providing a Ni single crystal, and for funding to the European Research Council under the European Union's Horizon 2020 Research and Innovation Programme (Grant No. 856538, project "3D MAGiC") (A.K., F.Z., R.D.-B.), to the Deutsche Forschungsgemeinschaft (Project-ID 405553726 TRR270) (A.K., R.D.-B.), to the China and Germany Postdoctoral Exchange Program 2019 from the Office of China Postdoctoral Council and the Helmholtz Centre (Grant No. ZD2019020) (D.K.), to the "111" project (DB18015) (D.K., L.W., X.H.), and to the Louisiana Board of Regents [Contract No. LEQSF(2020-23)-RD-A-32] (M.C.).

## Author contributions

X.H., A.K. and R.D.-B. designed the project. D.K. conducted FIB preparation and in situ TEM experiments. D.K., A.K. and F.Z. performed the analysis of the TEM results. M.C. performed the micromagnetic simulations. D.K., A.K., M.C. and R.D.-B. wrote the manuscript. D.K., L.W. and X.H. made the bimetallic deformation device. All authors contributed to the interpretation of the data and to writing of the manuscript text.

## Funding

## Competing interests

The authors declare no competing interests.
