## [Peer Review File · Nature Communications]

Reviewers' Comments:

Reviewer #1:

Remarks to the Author:

After careful consideration, I'm pleased to say that my concerns have been addressed and the article can be published as is. I appreciate the thoroughness of the authors in responding to my comments and revising the article accordingly. The changes made have significantly improved the clarity and rigor of the manuscript, and I believe it makes an important contribution to the field.

Overall, I find the study to be well-designed, the methodology to be appropriate, and the results to be compelling. The authors have also done an excellent job of contextualizing their findings within the broader literature.

Reviewer #2:

Remarks to the Author:

Review of manuscript "Direct observation of tensile-strain-induced nanoscale magnetic hardening" transferred to Nature Communications by Deli Kong et al.

Manuscript ID: NCOMMS-23-07182-T

I have read through the authors' revised manuscript, together with other two referees' remarks on their previous manuscript. Although I am almost satisfied with the authors' response to my previous comments and suggestions, I would like to request the authors one more brush-up of their manuscript before I recommend the manuscript for a publication. I believe that such a brush-up would raise the quality of the manuscript further to fulfill the standard of a high-impact journal like Nature Communications. The two major revisions and several minor revisions are listed below.

Major revisions

1. Abstract and the last paragraph of Introduction must be carefully revised stressing the significant advance of the present work that they are claiming in their response to Reviewer 3's comment #1. The novel point of the manuscript is the quantitative measurements of domain wall dynamics using a combination of off-axis electron holography and bimetallic deformation device, in addition to the dynamic observation by Lorentz TEM.
2. Fig. 1: To help the readers' comprehensions, I recommend the authors to merge Fig. s1 and Fig. 1 as a new Fig. 1. Refer to Fig. 1 of K. Shibata, et. al., Large anisotropic deformation of skyrmions in strained crystal, *Nature Nanotechnology*, 10, 589 (2015).

Minor revisions

1. Page 1, Abstract, line 21: "magnetic field" -> "magnetization" or "magnetic domain"
2. Page 2, line 37: ", which that lead" -> ", which lead"
3. Page 2, line 57: "in situ TEM" -> "in situ Lorentz TEM"
4. Page 3, line 77-82: It is strange the 2nd tensile cycle is described before mentioning the 1st tensile cycle. Explain the 1st cycle and "the starting point" determined by the 1st tensile cycle clearly at the beginning of the second paragraph.
5. Page 5, Fig. 2 legend: "C" and "D" -> "c" and "d"
6. Page 7, "Discussion": I think the content of "Plastic deformation" is not Discussion, but one of their Results. Instead, add Discussion before the summary. I think it is a good idea to include their responses to Reviewer 3's comment 2 and 3.
7. Page 7, Fig. 3: It is better to designate the stacking fault by a dotted line. I am afraid most readers are not able to identify the defect in these figures.

8. Page 8, line 221: Better to add a subtitle like "Effect of changing strain direction" here, as the following content is not "Plastic deformation".
9. Throughout the manuscript: Unified usages of technical terms such as "Fresnel defocus imaging" or "Lorentz TEM imaging" and "electron holography (EH)" or "Off-axis EH" are favorable.
10. Fig. s5 -> Fig. s1. Explain the 1st cycle and what is determined as "the starting point" by the 1st tensile cycle clearly in the figure legend.

Personal opinion: Page 8, line 199: "The stacking faults had no observable effect on the magnetic structure" -> By a careful examination of movie s2, it appears that the movement of magnetic walls is slightly affected by the stacking fault. In my opinion, magnetic domain walls are more or less influenced by structural defects.

End of Review

Response to Reviewer #2

We are very grateful to the referee for his or her thorough review and constructive suggestions. Below, we reply to all of the points, which are restated in italics. The corresponding changes are marked in red in the manuscript.

I have read through the authors' revised manuscript, together with other two referees' remarks on their previous manuscript. Although I am almost satisfied with the authors' response to my previous comments and suggestions, I would like to request the authors one more brush-up of their manuscript before I recommend the manuscript for a publication. I believe that such a brush-up would raise the quality of the manuscript further to fulfill the standard of a high-impact journal like Nature Communications. The two major revisions and several minor revisions are listed below.

Major revisions

1. Abstract and the last paragraph of Introduction must be carefully revised stressing the significant advance of the present work that they are claiming in their response to Reviewer 3's comment #1. The novel point of the manuscript is the quantitative measurements of domain wall dynamics using a combination of off-axis electron holography and bimetallic deformation device, in addition to the dynamic observation by Lorentz TEM.

Response: Thank you very much for the suggestion. We have revised the Abstract and Introduction, in order to communicate the significant advances of our work more clearly.

2. Fig. 1: To help the readers' comprehensions, I recommend the authors to merge Fig. s1 and Fig. 1 as a new Fig. 1. Refer to Fig. 1 of K. Shibata, et. al., Large anisotropic deformation of skyrmions in strained crystal, *Nature Nanotechnology*, 10, 589 (2015).

Response: Figure 1 and the previous Supplementary Fig. 1 are now combined into a new Fig. 1 in the main text.

Minor revisions

1. Page 1, Abstract, line 21: "magnetic field" -> "magnetization" or "magnetic domain"
2. Page 2, line 37: ", which that lead" -> ", which lead"
3. Page 2, line 57: "in situ TEM" -> "in situ Lorentz TEM"
4. Page 3, line 77-82: It is strange the 2nd tensile cycle is described before mentioning the 1st tensile cycle. Explain the 1st cycle and "the starting point" determined by the 1st tensile cycle clearly at the beginning of the second paragraph.
5. Page 5, Fig. 2 legend: "C" and "D" -> "c" and "d"
6. Page 7, "Discussion": I think the content of "Plastic deformation" is not Discussion, but one of their Results. Instead, add Discussion before the summary. I think it is a good idea to include their responses to Reviewer 3's comment 2 and 3.
7. Page 7, Fig. 3: It is better to designate the stacking fault by a dotted line. I am afraid most readers are not able to identify the defect in these figures.
8. Page 8, line 221: Better to add a subtitle like "Effect of changing strain direction" here, as the following content is not "Plastic deformation".
9. Throughout the manuscript: Unified usages of technical terms such as "Fresnel defocus imaging" or "Lorentz TEM imaging" and "electron holography (EH)" or "Off-axis EH" are favorable.
10. Fig. s5 -> Fig. s1. Explain the 1st cycle and what is determined as "the starting point" by the 1st tensile cycle clearly in the figure legend.

Thank you very much for the careful and helpful suggestions.

Response to points 1-3, 5, 7, 8, 9: The suggested changes to the text have been made.

Response to point 4: A sentence has been added in the suggested location, describing the role of the first tensile cycle.

Response to point 6: The text referring to plastic deformation is now labelled as a sub-section. The discussion has also been adapted.

Response to point 10: The caption to Supplementary Fig. 4 (formerly Fig. s5) has been extended, in order to include details about the 1st tensile cycle experiment.

Personal opinion: Page 8, line 199: "The stacking faults had no observable effect on the magnetic structure" -> By a careful examination of movie s2, it appears that the movement of magnetic walls is slightly affected by the stacking fault. In my opinion, magnetic domain walls are more or less influenced by structural defects.

Response: We agree with the reviewer that the presence of stacking faults may explain some minor observations in our experiments. However, we respectfully argue that the plastic deformation régime should be the subject of a separate extended study. Accordingly, we have changed the text in the present manuscript to “*The stacking faults had no **major** effect on the magnetic structure.*”